# Influence of Hospital Environmental Variables on Thermometric Measurements and Level of Concordance: A Cross-Sectional Descriptive Study

**DOI:** 10.3390/ijerph20054665

**Published:** 2023-03-06

**Authors:** Candelaria de la Merced Díaz-González, Noa Mateos-López, Milagros De la Rosa-Hormiga, Gloria Carballo-Hernández

**Affiliations:** 1Department of Nursing, Faculty of Health Sciences, University of Las Palmas de Gran Canaria, 35001 Las Palmas de Gran Canaria, Spain; 2Unit of Orthopaedic and Trauma Surgery, Hospital Insular de Gran Canaria, 35016 Las Palmas de Gran Canaria, Spain

**Keywords:** thermometers, body temperature, environmental factors, relative humidity, light, noise, temperature

## Abstract

During a pandemic, and given the need to quickly screen febrile and non-febrile humans, it is necessary to know the concordance between different thermometers (TMs) and understand how environmental factors influence the measurements made by these instruments. Objective: The objective of this study is to identify the potential influence of environmental factors on the measurements made by four different TMs and the concordance between these instruments in a hospital setting. Method: The study employed a cross-sectional observational methodology. The participants were patients who had been hospitalised in the traumatology unit. The variables were body temperature, room temperature, room relative humidity, light, and noise. The instruments used were a Non Contract Infrared TM, Axillary Electronic TM, Gallium TM, and Tympanic TM. A lux meter, a sound level meter, and a thermohygrometer measured the ambient variables. Results: The study sample included 288 participants. Weak significant relationships were found between noise and body temperature measured with Tympanic Infrared TM, r = −0.146 (*p* < 0.01) and likewise between environmental temperature and this same TM, r = 0.133 (*p* < 0.05). The concordance between the measurements made by the four different TMs showed an Intraclass Correlation Coefficient (ICC) of 0.479. Conclusions: The concordance between the four TMs was considered “fair”.

## 1. Introduction

Thermoregulation is a natural process that maintains body homeostasis and vital functions through the activation of central and peripheral nervous system mechanisms [1,2]. Body temperature (BT) is defined as an organism’s degree of heat conserved by the balance between the heat generated (thermogenesis) and heat lost (thermolysis) [3], expressed on a scale such as Celsius (°C), Kelvin (K), or Fahrenheit (°F) [4].

Thermogenesis covers the basal metabolic rate and muscle activity, among other measurements of thermolysis [4,5]; body heat is lost through radiation, conduction, convection, and evaporation. Many processes are involved in the maintenance of BT, but the heat exchange that occurs on the body’s surface is mainly mediated by the blood circulation [6]. BT depends on a multitude of factors (age, sex, state of health, level of consciousness, physical activity, emotions, location, and time of day, among others). In addition, when measuring this vital sign, it is necessary to select a TM [7].

Different body temperatures are produced when an imbalance exists between heat generation and loss. Fever occurs when an organism preserves the mechanisms of temperature regulation. Hyperthermia occurs when these mechanisms are out of control.

A normal BT is between 36.5 °C and 37.2 °C [8]. Therefore, core temperature values over 38.3 °C are considered to indicate a fever [9]. From a practical point of view, the BT will be >37.8 °C orally or >38.2 °C rectally [7]. These are values to consider during COVID-19 case detection. To detect any alteration in this vital sign, the BT measurements that are made must be accurate, valid, and reliable [10] in situations where the patient is, above all, outside of the normothermic range [11].

Thermometers (TMs) are necessary to determine BT. These devices allow the measurement of BT in different body areas, and include the outdated mercury TMs, Gallium TMs, digital TMs for oral, rectal, or axillary use, and tympanic and skin infrared TMs [12]. Importantly, BT is one of the most commonly measured vital signs in the control of a patient’s hemodynamic status.

The BT measurement procedure can affect the result, namely the choice of TM and the body area selected for measurement. Therefore, it is necessary to highlight rectal BT as the reference point for measurement. However, it is not widely used due to its invasiveness and limited practicality in the clinical setting [13]. This sometimes leads to a search for a fast and comfortable TM, prioritising performance over technique [10]. The measurement of BT at the tympanic level is defended based on its similarity to core temperature, due to the sharing of tympanic and hypothalamic vascularisation. Therefore, it is widely accepted as a valid area in which to measure BT [14] due to being painless, easy, and quick to read [15,16]. There are studies that show that these TMs have a tendency to underestimate or overestimate BT in situations where it is high [17]. Although it seems like an easy technique, meter training is important. Studies show differences in the measurements obtained by different technicians [18]. In addition, it is important to rule out the presence of cerumen in the ear canal. 

Environmental factors can also influence TMs. Such factors include temperature, light, noise, and humidity. Many studies [19,20,21,22] on thermography have failed to consider the inclusion of environmental factors that could increase or decrease similarity between the TMs used or lead to bias. For this reason, it is necessary to carry out a study that includes some of the environmental variables present in the patient’s room and assess their influences on different types of TMs, while also assessing the concordance.

Many types and models of TMs are currently used in the health field. Non contract Infrared TMs are in great demand, both in the health field and in the community (e.g., supermarkets, airports, nursing homes, and hospitals) due to the need to detect febrile individuals who may be infected with COVID-19.

This research aims to identify the possible influence of environmental factors on the measurements made by four different TMs and the concordance between these instruments in the hospital setting.

## 2. Materials and Methods

We followed the guidelines of Strengthening the Reporting of Observational Studies in Epidemiology (STROBE) [23].

### 2.1. Design and Context

This was a cross-sectional descriptive study. 

### 2.2. Setting

This study was developed in the Hospital Unit of Orthopaedic and Trauma Surgery (HUOTS) at the Hospital Insular de Gran Canaria (HIGC) over three months (from April to June 2021), allowing the recruitment of participants and their evaluation.

### 2.3. Participants

The participants in the study were patients admitted to HUOTS during the study period. The inclusion criteria were adults over 18 years of age, absence of alterations in their level of knowledge, absence of isolation prescription, absence of earwax in the ear canal, absence of skin diseases (for example, psoriasis), and written consent to participate in the study.

The sample collection procedure was as follows: (1) The researchers went to the patient’s room during their professional activity, limiting the number of people in the HUOTS. (2) The team introduced itself to the patient and delivered an information sheet and consent form, providing time to resolve any doubts that the patient may have. (3) Once the consent form was signed, the patient was considered a participant in the study and the measurements were carried out.

### 2.4. Variables

The variables are shown in Table 1.

### 2.5. Data Sources and Measurement

The evaluation tools used for the data collection were: (1) A Gallium TM. Measurement range: between 35.5 and 42 °C; resolution: 0.1 °C; accuracy +0.1 °C/−0.15 °C. (2) An Electronic Axillary TM MEDISANA^TM^ (newly acquired). Features: accurate oral, axillary, and rectal measurements of body temperature; acoustic fever alarm; automatic storage of the last memory result; automatic power off; digital LCD screen; flexible tip; °C to °F conversion; measurement results in 10 s; and certified medical device according to European regulations. (3) A Non-Contact Infrared Tympanic TM GENIUS 3^TM^ (property of the institution and calibrated by the electromedicine of that hospital, but no certificate available). Temperature accuracy of ±0.3 °C; probe tip ensures that it is also suitable for use in clinical settings; provides accurate and fast results; and temperature measurements are displayed on the large LCD screen 1–2 s after the scan button is pressed. (4) A Non-Contact Infrared TM BSX815 (new acquisition). Temperature range of 34.0–43.0 °C; accuracy of ±0.3 °C; resolution of 0.1 °C; environmental working temperatures of 10.0–40.0 °C; and relative humidity of 15–85%. (5) TESTO^TM^ 540 Lux Meter (accuracy ± 3 lux). (6) TESTO^TM^ 610 thermohygrometer. Temperature (accuracy of ± 0.5 °C) and humidity (accuracy of ±2.5%). (7) TESTO^TM^ 815 sound level meter (accuracy of ±1 dB).

The measurement procedure was as follows: (1) Environmental measurements (temperature, relative humidity, and light power) were performed, all of them in the same area (close to the patient, at a distance less than 1 m). (2) Patient measurements were taken. The electronic TM and the Gallium TM were placed in the same armpit. The Gallium TM was removed at 2 min and the electronic TM was removed when the completion beep sounded. The tympanic temperature was taken in the ear on the same side of the body as the selected axilla. Body temperature was taken on the forehead with a distance of 1–3 cm and, when the final sound signal played, it was recorded.

### 2.6. Study Size

The minimum sample size was 286 participants, based on a 50% response distribution, a 5% margin of error, and a 95% confidence interval, and considering the mean annual admissions to HUOTS of 1113 patients. 

### 2.7. Statistical Methods

Descriptive statistics were used to present the data: arithmetic mean (M), whose value determines the average level of a given variable, and standard deviation (SD), a statistical measure of scattering the results around the expected value. 

After the data collection and measurements were carried out, they were registered in a database created for this purpose, which was statistically analysed using the statistical software JASP 0.16.04 Apple Silicon (JASP Team, Amsterdam University, Amsterdam, The Netherlands) [24]. The following statistical analyses were carried out: descriptive statistical analysis and bivariate analysis—Pearson and Student’s *t*-test correlation. 

The study of concordance among the four TMs was performed using the Intraclass Correlation (ICC) [25,26,27] and the results were interpreted using the Landis–Koch criteria. These criteria consider concordance as “very good” if ICC > 0.91, “good” if ICC is from 0.71 to 0.90, “moderate” if ICC is 0.51–0.70, “fair” if ICC is 0.31–0.50, and “poor” if ICC is below 0.31. Possible systematic errors were detected using Bland–Altman plots [28].

### 2.8. Ethical Considerations

This study was authorised by the Dr. Negrín University Hospital Research Ethics Committee, with the code CEI 2020-327-1.

All participants received the participant information sheet, “informed consent”. Only those participants who signed the consent were included.

## 3. Results

### 3.1. Sample Characteristics

The sample obtained consisted of 312 participants, with a mean age of 68.63 years, an age range of 31–93 years, and a standard deviation (SD) of 15.78. All participants were Caucasian with skin phototype II–IV. The gender distribution of the sample was as follows: 57.7% (n = 180) male participants and 42.3% (n = 132) female participants. 

The bed location in the room was either next to the door or the window. A total of 51.9% (n = 162) were close to the window, far from the air conditioning system. A total of 48.1% (n = 150) of the participants were below the air conditioning system (next to the door).

### 3.2. Environmental Characteristics

Table 2 shows the characteristics of the environmental variables.

### 3.3. Descriptive Statistics

Table 3 shows the descriptive statistics of the BT measurements with four different TM. The data show higher means in the two infrared TMs.

### 3.4. Relationship between the Age and Gender of the Participants and Body Temperature

Table 4 shows the presence of a significant correlation between age and BT with all TMs, where the increase in age is correlated with a decrease in BT, and a positive correlation stands out in the relationship of BT with the Non-Contact Tympanic Infrared TM.

Table 5 shows the relationship between gender and the temperature measurement performed by Non-Contact Forehead Infrared TM (*p* < 0.001).

### 3.5. Relationship of Patient Bed Location to Body Temperature and Environmental Variables

The independent samples *t*-test (Student’s *t*-test), Shapiro–Wilk test of normality, and the test of equality of variance (Levane’s test) were used to assess the normality of the continuous quantitative variables and the categorical variables. The significant results suggest a deviation from normality.

After evaluating the equality of variances (Levane’s test) (*p* > 0.05), as shown in Table 6, we found no relationship between the location of the bed and the different BTs. In Table 7, it can be observed that something similar occurred. The categorical variable of bed location had no relationship with the environmental variables.

### 3.6. Correlations between Measurements Taken with the Four TMs

Table 8 shows the correlations between the BT measurements taken with the four different TMs, showing different combinations. There is a significant correlation in all cases, from low (see Table 8, rows 5 and 6) and moderate (rows 1, 2, and 3) to a strong correlation between the two TMs used at the axillary level, r = 0.739 (*p* < 0.001) (see row 4).

The ICC (see Table 9) provides a valid interpretation in the case of agreement. The Bland–Altman plot revealed both systematic and random errors. The ICC (0.479) shows a “fair” (0.31–0.50) agreement among the four TMs.

In Figure 1, Figure 2, Figure 3, Figure 4, Figure 5 and Figure 6, the means of two TMs are represented using Bland–Altman plots.

Figure 1, Figure 2, Figure 3, Figure 4, Figure 5 and Figure 6 show the measurements of two TMs against the difference of the measurements, with a limit of agreement of 95%.

The points are grouped around a “zero” line and the degree of dispersion, which is determined by the amplitude of the differences in the results between the TMs. The greater the degree of dispersion, the worse the agreement between the two TMs. The lines that coincide with a standard deviation below and above the zero mean were included, which represent the limits that we deemed acceptable between the two methods to consider good agreement. The Non-Contact Tympanic Infrared TM and Non-Contact Infrared TM (Forehead) (see Figure 3) and Axillary Electronic TM and Axillary Gallium TM (see Figure 4) are the TMs that present a better agreement in uniformity. Despite the agreement between the first two TMs, it is necessary to remember that both are the least accurate; hence, both are in agreement.

### 3.7. Correlation between Environmental Variables and Body Temperature

The correlation between the environmental variables and the temperatures measured by the four TMs are presented in Table 10, and two significant relationships were found: noise and environmental temperature present a weak relationship with body temperature as measured with the Non-Contact Infrared Tympanic TM, r = −0.146 (*p* < 0.01) and r = 0.133 (*p* < 0.05), respectively. 

## 4. Discussion

The sample obtained was representative of the study population (312 participants). The number of participants who had their bed located near the door and near the air conditioning system was 48.1%. The majority of the sample was located in the area near the window, the warmest area.

The mean age was 68.63 years, with an age range of 31–93 years. The mean is correlated with the measurements taken with all the TMs. BT decreased with age, an aspect that was measured with all TMs, except the Non-Contact Infrared TM, whose correlation with ageing was positive. The age of the participant (paediatric/adult) can influence the sensitivity in the measurements of the different TMs.

The highest percentage of the sample was male (57.7%). No relationship was found between this variable and the BT reported by the different TMs. However, the literature [29,30] shows a significant association between gender and BT, with females showing the highest BT due to responses to exogenous and endogenous heat gain and loss according to the phase of the menstrual cycle.

During the months of data collection (April to June), the environmental variables were: a mean humidity of 58.8%, a mean temperature of 23.0 °C, a mean luminosity of 81 Lux, and a mean noise of 62.1 dB. The provincial averages in those months are 64.5% humidity and 21.7 °C temperature [31]. 

Regarding environmental noise, the legislation in Spain (R.D. 1038/2012, of 6 July) [32] limits noise in the health field. The limit during the morning and afternoon is 40 dB, and the limit at night is 30 dB. At the time the measurements were taken (daytime), the minimum noise level was 38.1 dB, very close to the maximum allowed. In some cases, noise levels were double, with the maximum allowed at 84.1 dB. During the measurements, the researchers did not modify the variables. Therefore, factors such as the presence of noise from mobile phones, television, music, and conversations were respected.

Although our study found no relationship between ambient light and TM measurements, Piccinini et al. [33] found that the slightly incorrect use of Non-Contact Infrared TMs can leads to substantially unreliable measurements. Their results showed that measurements are influenced by light. Therefore, ambient radiation in the assessment room needs to be controlled by, for example, using constant artificial lights. The angle of inclination can lead to large discrepancies between subsequent estimates of BT, which is also demonstrated in normal lighting situations where no significant differences were found, except in environments where there was greater lighting [34]. 

The means of BT (Table 3) measured by the Non-Contact Tympanic Infrared TM and the Non-Contact Forehead Infrared TM were very similar (36.559 °C vs. 36.497 °C), a fact that was later revealed with a positive correlation in Table 6 (r = 0.352; *p* < 0.001). Furthermore, Figure 3 shows a low degree of dispersion, indicating a good agreement. Previous studies in the Community of the Canary Islands (Spain) [5] showed a higher BT, the difference being 1.3 ± 0.7 °C (38.5 vs. 37.2 °C). This indicates that they did not recommend this type of device due to the great difference in recorded temperature between the two devices. These two TMs can be influenced by different factors: in the case of Non-Contact Infrared TMs by the presence of cerumen (tympanic), sweat (evaporation decreases heat in the measurement area), low blood pressure levels, or vasoconstriction treatment (in both cases peripheral vascularization decreases), and the skin being thicker [35] in adults than in children (which causes a decrease in infrared radiation). Latman [36] conducted a review determining the possible causes of the relative lack of accuracy and reliability of Non-Contact Infrared Tympanic TM (among others), which also suggests combining these with an otoscope, although there is a range of normal anatomical variability in the ear canal in humans.

The means of the Axillary Electronic TM (35.795) and Axillary Gallium TM (35.769) were almost equal, which is reinforced by the significant strength of the correlation (r = 0.739; *p* < 0.001), and with a very low degree of dispersion (range between the limits), showing excellent agreement. Both coincide in contact with the skin, as well as in the same anatomical area, the axillary artery, which could be increased by improving this relationship. The same study cited above [5] presented a mean difference of 0.2 ± 0.4 °C (38 vs. 37.8 °C), recommending the regular use of these thermometers in clinical practice due to their accuracy. 

Other TMs present “moderate strength” concordance. This is true for the case of the Non-Contact Tympanic Infrared TM and Axillary Gallium TM (0.310) and the case of the Non-Contact Tympanic Infrared TM and Axillary Electronic TM (0.298). In both cases, there is a difference between the means (0.79 °C vs. 0.76 °C), which influences the plots (Figure 1 and Figure 2), where the range between the limits is greater than the previous TMs, indicating a lower agreement. Some authors have highlighted the poor performance of electronic axillary and tympanic thermometers compared to measurements of core body temperature [37]. Kameda [38] stated that a Tympanic Infrared TM can effectively approximate the core temperature and is quick and easy to use. Importantly, the frequent use of these instruments in clinical settings generates a significant amount of waste production, particularly of disposable plastic cannulas (caps). However, the study in [38] presented a rather small sample size and the correlation was surprisingly good, especially for Non-Contact TMs on the forehead. In most cases, this is not possible because the influencing factors (especially sweating in febrile but also non-febrile subjects), especially with elevated body temperatures, are too robust and cause large deviations. Additionally, most forehead TMs are cheap versions of non-contact TMs, which also have other problems (including detector instability, emissivity adjustment, thermometer case heating, and very wide viewing angle) that reduce their accuracy.

The TMs that showed a worse agreement but a significant correlation were the combination Non-Contact Infrared TM (Forehead) and Axillary Electronic TM or Axillary Gallium TM. The first has two sensors, one dedicated to the measurement of the skin temperature itself, while the other measures the environmental temperature, and based on the clinical equalization calculations, the BT is deduced, which is the result of the difference between these two measured values [39]. Studies [40] in this line have shown that the emissivity of the skin (the ratio between the radiation emitted by this surface and the radiation emitted by a black body at the same temperature) can be considered to be between 0.94 and 0.99; therefore, TMs should be adjusted accordingly. However, additional modern literature shows emissivity with a value of 0.98 [41]. This last value is used as an emissivity adjustment in these instruments, a value that is present and cannot be modified by the user, thus minimizing the source of uncertainty. These instruments account for both clinical bias and uncertainty. Stacey et al. [42] demonstrated that using six different models (versus an oral TM model) exceeded the accuracy claimed on their product labelling. One TM model (with the highest clinical bias) stood out as having 88% of the data outside the established accuracy and may not be precise enough to determine if the participant’s temperature exceeds the specific threshold of 38 °C. As in our case, Patel et al. [43] showed that a correlation between infrared and axillary thermometers is usually poor, which means that these devices are not interchangeable. Khan et al. [29] concluded that Non-Contact TM (versus Temporal Arterial TM) may not be a more accurate device for mass fever detection during a pandemic (low sensitivity of 16.13% for BT > 37.5 °C). This reinforces Tay et al.’s [44] finding of low sensitivity (29.4%) for BT > 37.5 °C compared to an oral TM. Whag et al.’s [45] study on Non-Contact Infrared TM and Infrared TMs stated that the latter can provide superior performance by being more accurate and, therefore, has a greater diagnostic efficiency. Subramanian et al. [46] showed that BT measurements on the forehead, when compared to the wrist, with a Non-Contact Infrared TM better reflect tympanic BT. Hayward et al. [47] found >1 °C variation in the two Non-Contact Infrared TMs, compared to Axillary Electronic and Non-Contact Tympanic TMs, in a primary care paediatric population. The limit of agreement considered acceptable in TM is ±0.5 °C [48], although the international standard for clinical TM must also be taken into account (ISO 80601-2-56:2017) [49]. In contrast, other authors [50] in a randomised controlled study showed that a Non-Contact Infrared TM was efficient, safe, and has the potential to improve patient satisfaction with medical care. Kameda N. [38] stated that Non-Contact Infrared TMs showed a good correlation with core temperature and the correlation coefficient was higher than those for temperatures measured with tympanic TMs and infrared axillary TMs. 

The ICC (0.479) between the four TMs showed a “fair” (0.31–0.50) agreement. We did not find previous studies that included the four TMs. However, there is evidence [5] that BT measurements with Non-Contact Infrared Tympanic TMs have a “moderate” agreement (0.55–0.58) when compared to each other in the left and right ear. This concordance improved to “very good” with a digital TM (0.90–0.97). We cannot generalise these concordances to all TMs. There are differences in technology, models, and brands that can influence their reliability. 

Regarding the influence of environmental variables on the measurements made by the four TMs, temperature (0.133; *p* < 0.05) and environmental noise (−0.146; *p* < 0.001) have a significant correlation with the BT measured exclusively by the Non-Contact Infrared Tympanic TM. The literature shows that air temperature and humidity vary greatly depending on the region and the season of the year. Therefore, temperate and/or humid climates produce an increase in BT. The normal response to this increase in temperature is sweat, which cools the body through evaporation. 

Other authors [5] also found a decrease in ambient temperature with the measurements made by the Non-Contact Infrared Tympanic TMs (<0.001). However, in the article in general, the author expressed a particularly negative opinion towards this type of TM.

Regarding noise, no articles have been found that relate it to temperature measurements with different instruments. Based on the fact that noise is a wave phenomenon through which mechanical energy is propagated in an elastic medium (air), it can be considered that this phenomenon could displace the beam of infrared light emitted by the non-contact thermometer, thus preventing a correct measurement of this area. It is necessary to continue investigating this variable.

Furthermore, the performance of TMs depends on the operator, in addition to the possible environmental variables. In the case of the Non-Contact Infrared TM, the distance between the thermometer and the skin can affect accuracy. Operator training is vital, in addition to other factors that have not been considered in this study. These include inflammatory skin conditions, breast cancer, systemic inflammatory diseases, septic shock, and the potential for wound healing [50]. The inclination of the light beam of these TMs can also influence the measurements [33].

On the other hand, due to the variation in values between the selected instruments, doctors and nurses must interpret the readings of the thermometers with caution. They must evaluate the patient alongside the data recorded in the Electronic Clinical Record (ECR). It must be further considered that there are ECRs that do not have a section to indicate the device used for said measurement. Because clinical guidelines are often based on specific fever thresholds, clinicians should interpret peripheral thermometer readings with caution, which must also be conducted immediately after the measurement. Once the data are stored in any database or registry, no one would be able to recover the causes of possible deviations because there are no additional data to allow further interpretation. 

### Limitations

Biases may have influenced the measurements in this study. Other environmental variables were not included in this study, beyond the participants’ variables. In addition, this study did not compare the methods included in it with a reference standard (core thermometry). However, a clinical TM (Axillary TM) was used, which already has an established relationship to core temperature and has acceptable [49] accuracy, but it was not calibrated by an accredited laboratory to provide traceable measurements.

This is due to the invasiveness of the process, cost, and lack of feasibility and appropriateness in a hospital unit. All participants were Caucasian with skin phototype II–IV, which limited the collection of data on dark skin types. 

## 5. Conclusions

The TMs with the best correlations were the Axillary Electronic TM and Axillary Gallium TM. The one with the worst correlation with other TMs was the Non-Contact Forehead Infrared TM. When comparing this instrument with the other TMs, it had the least correlation with them. There is a “fair” concordance between the four TMs studied. The environmental factors of temperature and environmental noise present a significant correlation with the BT measured exclusively by the Non-Contact Tympanic Infrared TM. The relationship between environmental temperature and BT has been shown in other investigations; however, there are not enough studies that support the relationship between noise and BT. It can be concluded that the sensor of Non-Contact Tympanic Infrared TMs may be more sensitive to environmental noise or the beam of light emitted by these TMs could have been displaced by noise waves, decreasing the area to be measured, an approach that must be studied in depth.

It is necessary to highlight the advantages of Non-Contact Infrared Forehead TMs in pandemic situations, where avoiding contact is essential, as well as speed and not generating waste, in addition to performing rapid screening among subjects with pyrexia and apyrexia. However, based on the results obtained in this study, these TMs reported the worst concordances with other TMs, being inaccurate in themselves. For this reason, taking precautions and conducting user training will not improve the measurements. In agreement with other authors [33], they have high rates of bias. On the other hand, users (for example, in airports and stores) must receive basic training in the use of these instruments, in addition to providing a suitable place to carry out the measurements, with controlled temperature, humidity, light, and noise. It is necessary to continue research on this topic and generate new evidence on environmental variables as well as other factors present in health settings to elucidate additional confounding factors. 

## Figures and Tables

**Figure 1 ijerph-20-04665-f001:**
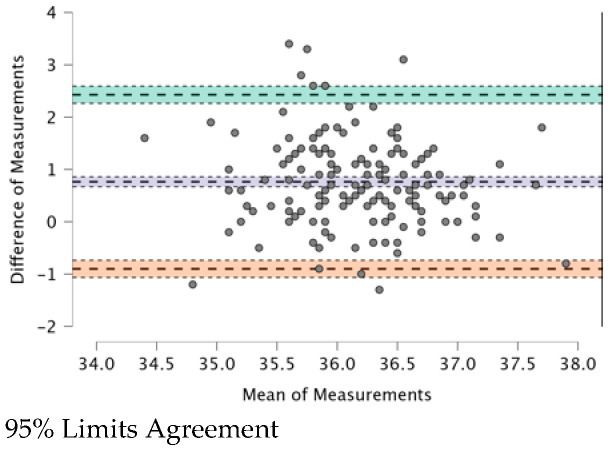
BT–Non-Contact Tympanic Infrared TM and BT–Axillar Electronic TM.

**Figure 2 ijerph-20-04665-f002:**
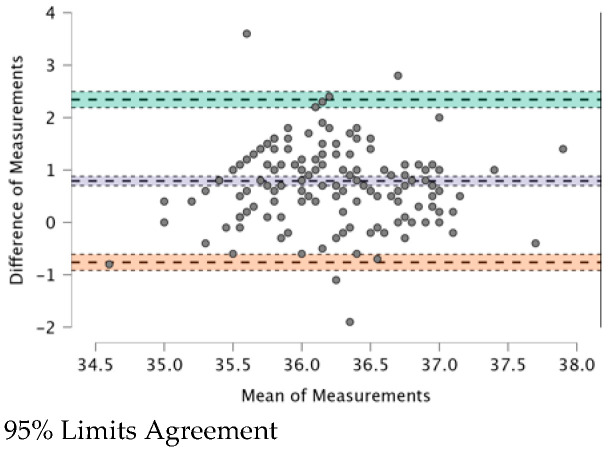
BT–Non-Contact Tympanic Infrared TM and BT–Axillar Gallium TM.

**Figure 3 ijerph-20-04665-f003:**
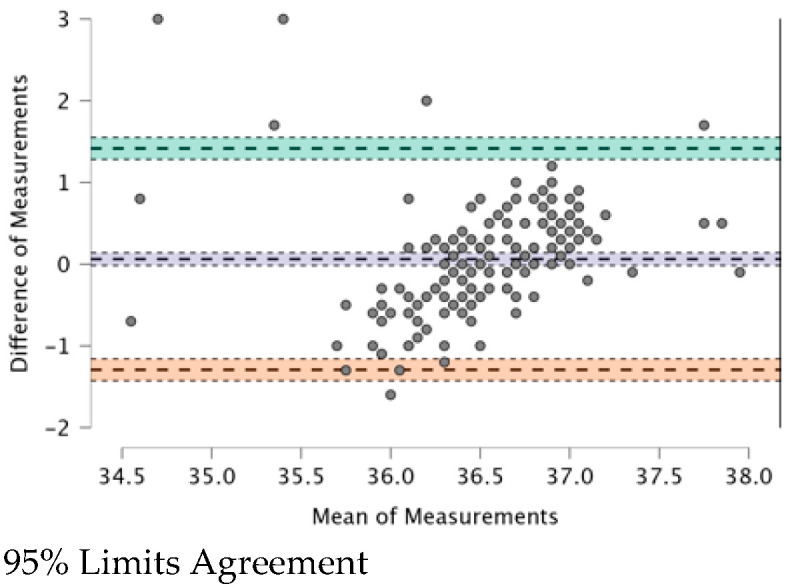
BT–Tympanic Infrared TM and BT–Non-Contact Infrared TM (Forehead).

**Figure 4 ijerph-20-04665-f004:**
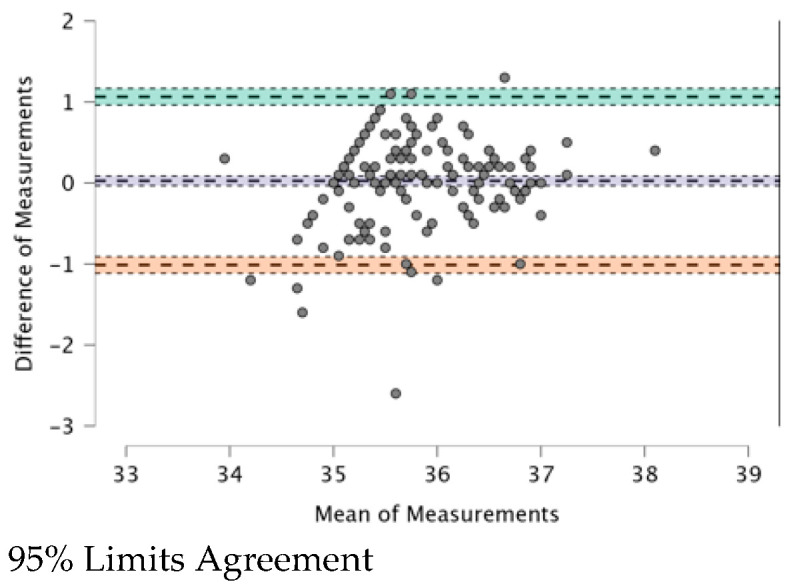
BT–Axillar Electronic TM and BT–Axillar Gallium TM.

**Figure 5 ijerph-20-04665-f005:**
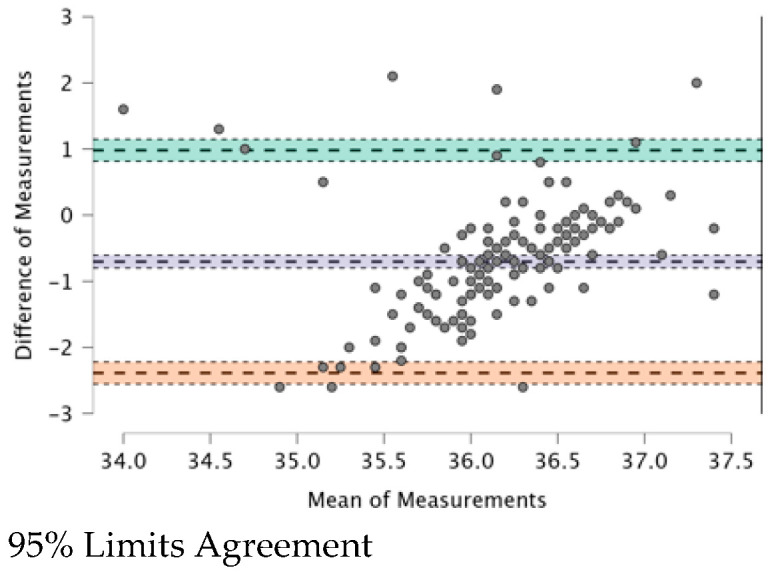
BT–Axillar Electronic TM and BT–Non-Contact Infrared TM (Forehead).

**Figure 6 ijerph-20-04665-f006:**
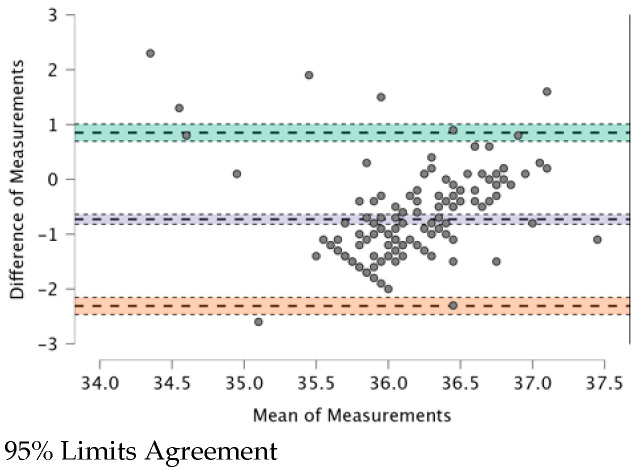
BT–Axillar Gallium TM and BT–Non-Contact Infrared TM (Forehead).

**Table 1 ijerph-20-04665-t001:** Independent and dependent variables.

Variables	Description
Independent Variables	
Age	Age of the participant in years
Gender	Male or female
Location	Location of the bed in the room: door (below the air conditioning outlet) or window (far from the emitter).
Temperature	Environmental Temperature. Unit: Celsius = °C.
Humidity	Environmental Relative Humidity. Unit: percentage = %.
Light	Environmental Light. Unit: lumen = Lux.
Noise	Environmental Noise. Unit: Decibels = dB.
Dependent variables	
Axillary Body Temperature	Body temperature measured with a Gallium Thermometer.
Tympanic Body Temperature	Body temperature measured with a Non-Contact Tympanic Infrared Thermometer.
Forehead Body Temperature	Body temperature measured with a Non-Contact Forehead Infrared Thermometer.
Axillary Body Temperature 2	Body temperature measured with an Electronic Thermometer.

**Table 2 ijerph-20-04665-t002:** Environmental variables.

	Environmental Humidity (%)	Environmental Temperature (°C)	Environmental Light (Lux)	Environmental Noise (Db)
Valid	312	312	312	312
Missing	0	0	0	0
Mean	58.878	23.032	81.037	62.148
Std. Deviation	5.325	1.109	30.176	12.708
Minimum	44.400	20.000	11.000	38.100
Maximum	75.600	24.700	172.000	84.100

**Table 3 ijerph-20-04665-t003:** Body temperature.

	BT–Tympanic Infrared TM	BT–Axillary Electronic TM	BT–Axillary Gallium TM	BT–Non-Contact Infrared TM (Forehead)
Valid	312	312	312	312
Missing	0	0	0	0
Mean	36.559	35.795	35.769	36.497
Std. Deviation	0.660	0.766	0.688	0.542
Minimum	34.200	33.600	33.800	33.200
Maximum	38.600	38.300	37.900	38.000

TM = Thermometer. BT = Body temperature.

**Table 4 ijerph-20-04665-t004:** Correlation of age and body temperature.

		BT–Tympanic Infrared TM	BT–Axillary Electronic TM	BT–Axillary Gallium TM	BT–Non-Contact Infrared TM (Forehead)
Age	Pearson’s r *p*-value	0.166 ** 0.003	−0.156 ** 0.006	−0.147 ** 0.009	−0.403 *** <0.001

** *p* < 0.01, and *** *p* < 0.001. BT = Body temperature.

**Table 5 ijerph-20-04665-t005:** Independent samples *t*-test: gender and body temperature.

Independent Samples *t*-Test							
	T	Df	*p*	Mean Difference	SE Difference	95% CI Mean Difference	
						Lower	Upper
BT–Non-Contact Tympanic Infrared TM	0.275	310	0.784	0.021	0.076	−0.128	0.170
BT–Axillary Electronic TM	−0.490	310	0.625	−0.043	0.088	−0.216	0.130
BT–Axillary Gallium TM	−0.791	310	0.430	−0.062	0.079	−0.217	0.093
BT–Non -Contact Infrared TM (forehead)	5.398	310	<0.001 ***	0.321	0.060	0.204	0.438

Student’s *t*-test. *** *p* < 0.001. BT = Body temperature.

**Table 6 ijerph-20-04665-t006:** Independent samples *t*-test: bed location and body temperature.

	T	Df	*p*	Mean Difference	SE Difference	95% CI Mean Difference	
						Lower	Upper
BT–Non-Contact Tympanic Infrared TM	0.301	310	0.764	0.023	0.075	−0.125	0.170
BT–Axillary Electronic TM	1.031	310	0.303	0.089	0.087	−0.081	0.260
BT–Axillary Gallium TM	1.305	310	0.193	0.102	0.078	−0.052	0.255
BT–Non-Contact Infrared TM (forehead)	−0.839	310	0.402	−0.052	0.061	−0.173	0.069

Student’s *t*-test. TM = Thermometer. BT = Body temperature.

**Table 7 ijerph-20-04665-t007:** Independent samples *t*-test: bed location and environmental variables.

	T	Df	*p*	Mean Difference	SE Difference	95% CI Mean Difference	
						Lower	Upper
BT–Non-Contact Tympanic Infrared TM	−0.879	310	0.380	−0.531	0.604	−1.718	0.657
BT–Axillary Electronic TM	−0.633	310	0.527	−0.080	0.126	−0.327	0.168
BT–Axillary Gallium TM	0.264	310	0.792	0.905	3.424	−5.833	7.644
BT–Non-Contact Infrared TM (forehead)	1.282	310	0.201	1.844	1.438	−0.987	4.674

Student’s *t*-test. TM = Thermometer. BT = Body temperature.

**Table 8 ijerph-20-04665-t008:** Correlations: body temperature and different thermometers.

		Pearson’s r	*p*-Value
BT–Non-Contact Tympanic Infrared TM	CT–Non-Contact Axillary Electronic TM	0.298 ***	<0.001
2. BT–Non-Contact Tympanic Infrared TM	CT–Axillary Gallium TM	0.310 ***	<0.001
3. BT–Non-Contact Tympanic Infrared TM	CT–Non-Contact Infrared TM (forehead)	0.352 ***	<0.001
4. BT–Axillary Electronic TM	CT–Axillary Gallium TM	0.739 ***	<0.001
5. BT–Axillary Electronic TM	CT–Non-Contact Infrared TM (forehead)	0.172 **	0.002
6. BT–Axillary Gallium TM	CT–Non-Contact Infrared TM (forehead)	0.156 **	0.006

** *p* < 0.01, *** *p* < 0.001. TM = Thermometer. BT = Body temperature.

**Table 9 ijerph-20-04665-t009:** Interclass correlation among the four thermometers.

	Point Estimate	Lower 95% CI	Upper 95% CI
Thermometers	0.479	0.378	0.567

Note. A total of 312 participants and 4 raters/measurements. ICC type as referenced by Shrout and Fleiss (1979).

**Table 10 ijerph-20-04665-t010:** Correlations: environmental variables and body temperature with different thermometers.

		BT–Non-Contact Tympanic Infrared TM	BT–Axillary Electronic TM	BT–Axillary Gallium TM	BT–Non-Contact Infrared TM (Forehead)
Environmental Humidity, %	Pearson’s r *p*-value	0.038 0.502	−0.082 0.151	−0.072 0.202	−0.029 0.610
Environmental Temperature, °C	Pearson’s r *p*-value	0.133 * 0.019	0.045 0.431	−0.043 0.447	0.070 0.221
Environmental Light, Lux	Pearson’s r *p*-value	−0.050 0.375	0.041 0.466	0.031 0.584	−0.020 0.725
Environmental Noise, Db	Pearson’s r *p*-value	−0.146 ** 0.010	−0.081 0.152	−0.073 0.197	−1.127 × 10^−4^ 0.998

* *p* < 0.05, ** *p* < 0.01. TM = Thermometer. BT = Body temperature.

## Data Availability

The data that support the findings of this study are available from the corresponding author (Candelaria de la Merced Díaz-González on behalf of all authors) upon reasonable request.

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
