# Peer review of "Influence of Hospital Environmental Variables on Thermometric Measurements and Level of Concordance: A Cross-Sectional Descriptive Study"

_ijerph, 2023, doi:10.3390/ijerph20054665_

Round 1

Reviewer 1 Report

Comments of the referee:

General comment:

The topic is important for scientific, professional and also common audience. Measurement of humany body temperature has many dilemmas. Many of these were described in the article, which authors should consults because it is related directly to this article as well https://www.tandfonline.com/doi/full/10.1080/10739140903149061. The problem of body temperature measurement is far from trivial when predetermined accuracy must be met and/or decisions taken in the case of body temperature measurement. A more recent study was published with the most important question: Is current body temperature measurement practice fit-for-purpose?

https://www.tandfonline.com/doi/full/10.1080/03091902.2021.1873441

The study is interesting, various methods were used for data analysis and evaluation of results. The problem is a selection of statistical methods. Namely, calculation of correlation between two variables may be questionable, if there are other variables present, which also influence measurement results. Methods and materials are described but need some minor improvements to provide sufficient details to allow other researchers to reproduce the results or conduct further research on a similar topic. Figures and tables appropriately present the results. Conclusions are not very clear and potentially misleading because for the non contact forehead TM it states the worst concordance with other TMs and still suggest that this TM should be used with caution along with training of users. If the instrument is inaccurate itself, no caution or training could help to obtain accurate results!

Thorough reference list with no self-citation are supporting information well. With all due respect to Spanich language, there are 12 references out of 45, which might not be easily understandable to majority of readers.

The article has some novelty and originality in methodological approach, but conclusions are rather indecisive and potentially misleading as mentioned above regarding the non contact forehead TM. It provides some progress in the field, and might be considered for publication in the journal after major revisions are made based on specific comments.

Specific comments

C1: Abstract (line 19, 80, 111, 400, 405, 408 and elsewhere)

The instruments used were a No Contact Infrared TM

This should read “Non Contract Infrared TM…” On top of that it is a forehead thermometer and that should be clearly stated as well.

C2: Abstract (line 20)

thermohydrometer” should be replaced by “thermohygrometer”

C3: Abstract (lines 22-25)

Significant relationships were found: Noise and environmental temperature have a weak relationship with body temperature, measurements by the Tympanic Infrared TM r= -0.146 (p < .01) and r=0.133 (p < .05), and Intraclass Correlation Coefficient (ICC) (0.479). Conclusions. The ICC between the 4 TMs was considered "fair".

The statements in the last few sentences of the abstract are not very clear and understandable. The expression could and should be improved.

C4: Introduction (lines 31-32)

The introduction should be succinct, with no subheadings. Limited figures may be included only if they are truly introductory, and contain no new results.

This part of introduction remained by mistake and should be removed.

C5: Introduction (lines 39-40)

body heat is lost through radiation, driving, convection, and evaporation.

Heat flux is transfered in 4 different ways: radiation, convection, conduction and phase transition. What did you mean by “driving”? This should probably be replaced by “conduction”.

C6: Introduction (line 42)

Blood circulation is an important aspect of the measurement of heat. BT]

The part “BT]” should be removed? Or should be removed the part “of heat”?

Regarding the meaning of this sentence, it is a bit confusing. Blood circulation is definitely important in a way that it influences the body temperature. We don’t measure heat at all, at least not in this case. Rephrase!

C7: Introduction (lines 42-43)

BT, depends on the choice of a thermometer (TM), measurement area and time of day [7].

BT depends on the time of a day definitely. It depends also on blood circulation, physical activity, menstrual cycle in women, digestion, health status, etc. Measured temperature depends on the coice of a thermometer and measurement site, but BT itself does not depend on this!

C8: Introduction (line 47)

What does “central temperature values” mean? Did you mean core temperature? That is the term used in human body temperature measurement. Please, consider the statement and make it uniformly throughout the article. 

C9: Introduction (lines 52-54)

The BT measurements that are made must be accurate, valid and reliable [10] in situations where the patient is, above all, outside the normothermic range [11].

This is a duplicated sentence to the previous one and should be removed.

C10: Matherials and methods (line 111)

Body Temperature with Thermometer de Gallium. (Gallium Thermometer)

C11: Matherials and methods (lines 114-115)

The evaluation tools used for data collection were: (1) A Gallium TM: Measurement range: between 35.5 and 42ºC, resolution: 0.1°C, measurement precision: +0.1oC / - 0.15oC.

Correct the following throughout the article: the value and the unit must be written separately, e.g. 42 °C. For temperature in degrees Celsius the unit and symbol is °C (not 0.1oC).

C12: Matherials and methods (lines 120-121)

A GENIUS 3TM Infrared Tympanic TM (institution calibrated by electromedicine): temperature accuracy of +/- 0.1 °C ...

What does it mean: institution calibrated by electromedicine? Do you have a calibration certificate for this tympanic thermometer? Temperature accuracy for this thermometer is not
± 0,1 °C, not even in its specifications. Where does this information come from? The best calibration laboratories in the world, specialized for calibration of tympanic thermometers could not achieve better accuracy than ±0,2 °C!!!
What about the calibration status of other thermometers, expecially the non contact forehead thermometer? Were they calibrated? By whom? Did you receive the calibration certificates with stated corrections and the measurement uncertainty and what were the values?

C13: Materials and methods (line 124)

An Infrared TM BSX815

Here is missing at least the statement that this is a non contact thermometer. More important for the whole article is that also Genius thermometer is a non contact thermometer. So how could a reader know which thermometer is in question? The solution is additional statement which indicates the measurement site. For example non contact ear (tympanic) thermometer, non contact forehead thermometer. This should be uniformly stated throughout the article where many times is written “Front Body Temperature” but should be written “Forehead”. There is no front thermometer!

C14: Results (lines 172-174)

This section may be divided by subheadings. It should provide a concise and precise description of the experimental results, their interpretation, as well as the experimental conclusions that can be drawn.

This part of results remained by mistake and should be removed.

C15: Results (line 181)

Characteristics of were body temperature

This title is very confusing, rephrase!

C16: Results (line 191)

Table 4 Correlations. Age and Body Temperature.

What are conclusions based on the data in Table 4, especially regarding the only positive correlation of Tympanic Infrated TM?

C17: Results (line 192 and 195)

CT= Corporal Temperature

What is the Corporal Temperature?

C18: Results (line 194)

Table 5. Independent Samples T-Test. Gender and Body Temperature.

What are conclusions based on the data in Table 5?

C19: Results (lines 214-220)

Relationship between environmental variables and body temperature

Even without statistics it was expected that the best correlation will be at the same measurement site (axilla) and the worst between forehead and axilla. But the question is the following, what do you suggest regarding the influence of environmental temperature on body temperature measured with different thermometers? Further question is, what kind of correlation is calculated here; between environmental variables and body temperature (says the title 3.6)?

C20: Results (lines 226-239)

Figures 1-6, the means of two TMs are represented using Bland-Altman Plots

The Tympanic Infrared - TM and No Contact Infrared TM - Front (see Figure 3) and Axillar Electronic TM & Axillary Gallium TM (see Figure 4) are the TMs that present a better uniformity agreement.

What does it help, if we have a good agreement between tympanic and forehead thermometer, if both are less accurate than axillary (check the spelling in the above sentence and elsewhere) thermometers, especially bad results are always with the forehead thermometer, which is practically useless.

C21: Results (lines 246-248)

Table 10. Correlations. Environmental Variables and Body Temperature with Different Thermometers.

... finding two significant relationships: noise and environmental temperature present a weak relationship with the body temperature emitted by the Infrared Tympanic TM r= -0.146 (p < .01) and r=0.133 (p < .05).

The authors found only two statistically significant relations. How do you comment the influence of environmental variables on the body temperature measurements with different thermometers? Similar to the comment C19, it is a bit confusing that the section under the title 3.6 only relates to environmental variables in Table 10, if I understand correctly? Maybe authors should consider more titles with descriptive message.

C22: Discussion (lines 252-253)

The results and discussion may be presented separately, or in one combined section, and may optionally be divided into headed subsections.

What is the purpose of this sentence?

C23: Discussion (lines 293-294)

Both devices are imprecise because many factors can influence the measurement (presence of ear wax, increased blood pressure, sweat, or skin type).

While presence of ear wax and sweat incluence the measurement of non contact TM, please state the references how the increased blood pressure or skin type influence the same measurements.

C24: Discussion (lines 318-325)

Studies in this line have shown that the emissivity of the skin (the ratio between the radiation emitted by this surface and the radiation emitted by a black body at the same temperature) can be considered between 0.94 and 0.99, so the TMs must be adjusted accordingly. Any difference between the actual skin emissivity value and the setting emissivity is a source of uncertainty. The uncertainty is far from negligible (a few tenths of a degree Celsius) and depends on the ambient temperature. Indeed, the further away the skin is from ambient temperature, the greater the uncertainty [31].

While it is true that measurements of the skin emissivity in literature show the different values between 0.94 and 0.98, majority of these measurements are rather old. Latest measurements in most of the cases report the value 0.98, which is used for instrumental emissivity setting in those TMs, which have this feature. Most of the clinical non contact TMs have a pre-set value of skin emissivity and a user neither not able to change it, neither it is possible to display it. Further comment, the ambient temperature in a controlled environment does not have a substantial influence on the temperature measurement with non contact TMs.

C25: Discussion (lines 346-347)

This indicates that it TMs exceeds the limits of agreement of ±0.5°C, which is commonly believed to be acceptable.

Please read the above sentence and rephrase it, so that it will be understandable. Authors should also consider the international standard for clinical thermometers and make a reference to it ISO 80601-2-56:2017 Medical electrical equipment — Part 2-56: Particular requirements for basic safety and essential performance of clinical thermometers for body temperature measurement. In this standard there are stated the limits of agreement, as the authors call it. Generally, this expression is related to accuracy.

C26: Discussion (lines 350-353)

Kameda N. [36] states that No Contact Infrared TMs can effectively approximate core temperature and are quick and easy to use. They show a good correlation with core temperature and the correlation coefficient was higher than those for temperatures measured with tympanic thermometers and infrared axillary thermometers.

Kameda reference is 39 not 36 (the same applies for line 309). Kameda's really presented a rather small sample number and the correlation was surprisingly good, especially for the forehead non contact TM. In majority of cases that is not possible because the influence factors (especially sweat with febrile subjects but also non febrile), especially with elevated body temperatures are to strong and cause high deviations. Not to mention that most of forehead thermometers are cheap versions of non contact thermometers, which have also other problems (detector instability, emissivity setting, heating of the thermometer case, very wide viewing angle, etc.) that lower their accuracy.

C27: Discussion (lines 367-375)

Regarding environmental noise, Latman [44] conducted a review determining the possible causes of the relative lack of precision and reliability of Infrared Tympanic TM (among others). Latman [44] carried out a review determining the possible causes of the relative lack of precision and reliability of the Infrared Tympanic TM (among others), also suggests combining these with an otoscope, although there is a range of normal anatomical variability of the canal in humans. Other authors [5] also found a decrease in ambient temperature with the measurements made by the Infrared Tympanic TMs (< 0.001). Regarding noise, no documents have been found that relate it to temperature measurements with different instruments.

In the reference 44 I found nothing related to the noise. Am I missing something or some other reference should be mentioned?

In the doctorate thesis (reference 5) there is mentioned several times that non contact infrared thermometers should be used with a great causion because their results exhibit the largest deviations. Only the best thermometers could be used with acceptable accuracy but in general the users don't know which thermometers are accurate enough because all manufacturers claim the same accuracy. On top of that practically no country requires calibration of any clinical thermometer by an accredited laboratory. It is a free market and all sort of clinical thermometers with very different quality (accuracy) are sold.

Particularly negative opinion is expressed towards non contact forehead thermometers and that should be stressed also in this article.

C28: Discussion (lines 383-389)

On the other hand, due to the variation in values between the selected instruments, doctors and nurses must interpret the readings of the thermometers with caution. They must evaluate the patient alongside the data recorded in the Electronic Clinic Record (ECR). It must be further considered that there are ECRs that do not have a section to indicate the device used for said measurement. Because clinical guidelines are often based on specific fever thresholds, clinicians should interpret peripheral thermometer readings with caution, which was not considered in this study.

It is true that readings of clinical thermometers should be interpreted with caution but that must be done immediatelly after the measurement. Once the data is stored to any database or record, nobody could retrieve the causes of possible deviations because there are no additional data that would enable later interpretation.

C29: Discussion, limitations (lines 392-395)

In addition, this method comparison study does not compare the methods included in it with a reference standard (central thermometry). This is due to the invasive process, cost, and lack of feasibility and appropriateness in a hospital unit.

In the clinical study of clinical thermometers it is not necessary to have the reference thermometer, which measures temperature in one of the »golden« spots (pulmonary artery, urinary bladder, esophagus, tympanic membrane). The second option is to use a reference thermometer, which is a clinical thermometer that already has an established relation to the core temperature and has an acceptable accuracy (see ISO 80601-2-56:2017). Axillary thermometer is definitely one of those. But any such »secondary« reference must be calibrated first by an accredited laboratory to provide traceable measurements.

C30: Conclusions (lines 403-404)

The environmental factors of temperature and environmental noise present a significant correlation with the BT measured exclusively by the Infrared Tympanic TM.

The influence of environmental temperature on the measured temperature with some thermometers is well known but the influence of noise is not. How do you know that noise has the influence on the measured temperature? Namely, the sensor in different thermometers are not sensitive to noise, at least not at the detectable level? The same applies to light, if that is not close to the measurement site and radiates with a high energy. Could you support the statements by references-peer reviewed articles?

Author Response

Dear reviewer,

We are very grateful for the time dedicated to our document and for the valuable contributions made.

Tell him that we have made the modifications, and following his indications we will send it to the English Edition Service of MDPI.

We will resubmit the document shortly.

Greetings

Reviewer 2 Report

I believe that the main results reported in the manuscript by Diaz Gonzalez et al are of wide interest for the scientific community. The paper seems well structured and presents a solid statistical methodology. I have only one suggestion to further improve the results reported in this paper. Since the Authors considered several parameters and a high number of patients, I suggest them to investigate all their data using a principal component analysis. In this way, they can identify some clusters of their parameters and at the same time they can further support their previous results. Then, please rephrase the title of subsection 3.3. In addition, I also believe that the weak point of this paper is that it needs a full linguistic revision because the level of the English is very low and several sentences should be rephrased. I understand that the Authors are not English native speakers, but I suggest them to perform a linguistic revision.

Author Response

Dear Reviewer,

Thank you very much for your contributions. We will make the suggested changes and within 10 days we will resend the document, but before that, we will send the document to "editing service".

Greetings

Reviewer 3 Report

The manuscript ‘Influence of Hospital Environmental Variables on Thermometric Measurements and Level of Concordance: A Cross-Sectional Descriptive Study’ presented the environmental factors' potential influence on the measurements made by 4 different thermometers, including the concordance between these instruments in a hospital setting. It seems like a good experimental report with more detailed data and results.

However, it is difficult to find any scientific highlights and new technological innovations in the article. In the sections of the abstract, the introduction and the conclusion, the content also appears confused and the writing illogical in some sense. In this view, I think the paper cannot be published.

Author Response

Dear reviewer,

We are very grateful for the time dedicated to our document and for the valuable contributions made.

Tell him that we have made the modifications, and following his indications, we will send it to the English Edition Service of MDPI.

We will resubmit the document shortly.

Greetings

Round 2

Reviewer 1 Report

One last thing that authors MUST correct is the proper use of terms precision and accuracy. You use many times the term precision (lines 110, 301, 312, 320, 411), while in all these case term accuracy (or accurate) shall be used. Authors can find themselves explanation of the difference but bare in mind, an instrument may be very precise but that does not mean automatically that is is accurate.

Author Response

Dear Reviewer 1
Thank you again for your contributions and help, which has allowed us to substantially improve our manuscript.
Indeed, precision is the detail with which an instrument or procedure can measure a variable, while accuracy is how close this measurement is to the true value. The problem is the translation in Spanish, both can be translated as "precision".
We have already replaced the terms in the manuscript (blue color).
Greeting

Reviewer 3 Report

I still think the paper lacks novelty. It is difficult to find any scientific highlights and new technological innovations in the article. In this view, I think the paper cannot be published.

Author Response

Dear Reviewer 3
Thank you again for the time spent reviewing our manuscript and your contributions. We are sorry that you do not consider it of interest to publish.
Greeting